



# Spatio-temporal encoding by quadratic gradients in magnetic resonance imaging

Sina Marhabaie[1], Geoffrey Bodenhausen[1], Philippe Pelupessy[1]

[1]Laboratoire des biomolécules, LBM, Département de chimie, École normale supérieure, PSL University, Sorbonne
Université, CNRS, 75005 Paris, France

*Correspondence to*: Sina Marhabaie (sina.marhabaie@u-pem.fr)

**Abstract.** SPatio-temporal ENcoding (SPEN) MRI is a non-Fourier imaging technique that encodes the spatial information in such a way that there is a one-to-one correspondence between the signal intensity as a function of time and the spin density at the corresponding position. In current spatio-temporal encoding methods imparting a quadratic phase—that is the phase of the
transverse magnetization depends as a quadratic function of the spatial coordinates—onto the transverse magnetization is the first crucial step. Usually, this is achieved by simultaneous application of a frequency-swept (chirp) pulse and a linear magnetic field gradient. In this work, we show that it can be advantageous to use quadratic encoding gradients for this purpose. By avoiding chirp pulses one can achieve much smaller specific absorption rates (SARs), and shorter echo times (TEs), while the spatial resolution, the field of view (FOV), and the signal-to-noise ratio (SNR) are the same as in SPEN if one uses similar
parameters. In addition, the proposed sequence can readily be used for multi-slice applications.

## 1 Introduction

Most conventional MRI methods are based on Fourier transformations: $k$-space data points are sampled in all directions, and the full $k$-space is subjected to a two or three dimensional Fourier transformation to produce the desired image (Brown et al., 2014). This scheme is currently the most common in MRI, though other methods exist. Among alternative methods, spatial
encoding, also known as spatio-temporal encoding or SPEN (Ben-Eliezer et al., 2014; Schmidt et al., 2014) is based on the sequential excitation of the spins of the phantom, and sequential detection of the signals. Since signal detection is sequential in spatio-temporal encoding, it is sufficient to arrange the signals in chronological order of excitation to reconstruct an image, and there is no need for a Fourier transformation in the spatio-temporal encoding direction. So far, a variety of spatio-temporal encoding techniques have been introduced (Ben-Eliezer et al., 2010a, 2010b, 2012, 2014; Chamberlain et al., 2007; Ciobanu
et al., 2015; Goerke et al., 2011; Marhabaie et al., 2017; Ryu et al., 2019; Schmidt et al., 2014; Zhang et al., 2016a, 2016b). In most methods, spatial encoding is achieved by simultaneous application of a linear gradient and a frequency-swept (chirp) pulse. This gives rise to a quadratic phase profile (Pipe, 1995, 1996), which has a de-phasing effect on the MR signal of all voxels  except for those that lie near the vertex (the flat region) of the quadratic phase profile. In order to obtain an image from





the sample, the position of this vertex can be shifted by linear gradients. The signal acquired in this manner resembles the spin
density, and there is no need for any Fourier transformation to produce an image.

To obtain an acceptable resolution and to suppress distortions due to field inhomogeneities, chirp pulses with large bandwidths are required. Such pulses lead to a considerable increase of the SAR of spatio-temporal encoding pulse sequences. The situation is exacerbated in multi-slice experiments or any sequences that uses 180˚ chirp pulses (Ciobanu et al., 2015; Schmidt et al., 2014; Zhang et al., 2016a, 2016b), because these pulses are associated with much higher SAR than 90˚ chirp
pulses.

In order to obtain acceptable resolution while keeping the SAR within a reasonable limit, one has to use long chirp pulses. This will increase the distortions due $B_0$ inhomogeneities and prolong the minimum echo time TE of the sequence with concomitant signal losses. In addition, multi-slice experiments that use 180° chirp pulses cause problems due to the fact that these pulses excite the spins of the whole object and are not limited to one slice.

The above-mentioned problems can be avoided by generating a quadratic phase profile of the magnetization without resorting to *rf* pulses, and by using a gradient coil that produces a quadratic magnetic field profile. To give a proof of concept in this paper, we will use the shim coil of our magnet as a source of quadratic encoding gradients, and we show that quadratic encoding gradients indeed offer an attractive alternative to the use of chirp pulses in spatio-temporal encoding MRI techniques. It worth mentioning that in the alternative approach presented here, the inherent properties of the method do not change. This
means that traditional SPEN sequences (using chirp pulses) have theoretically the same properties as the modified sequences described in this work, provided that they have the same coefficient $\beta = 2\pi\gamma\alpha\tau$ of the quadratic term, where α is the amplitude of the quadratic encoding gradient and τ is its duration. For example, traditional and modified SPEN sequences should have the same resilience with respect to inhomogeneous magnetic fields (an advantage that is shared by both methods), but they suffer from the same loss in SNR with respect to *k*-encoding methods (a common disadvantage of both approaches).

Over the past years, many studies have been conducted with non-linear encoding gradients. Different facets of non-linear encoding fields in MRI have been discussed extensively, and suitable pulse sequences and image reconstruction methods have been published (Galiana et al., 2012a, 2012b; Gallichan et al., 2011; Hennig et al., 2008; Layton et al., 2012, 2013; Schultz, 2013; Schultz et al., 2010; Stockmann et al., 2010, 2013; Wu et al., 1994; Xu et al., 2013). To the best of our knowledge however, quadratic encoding gradients have not yet been applied to spatio-temporal encoding (SPEN) methods that normally
use chirp pulses.

## 2  Theory

The pulse sequences used for multi-shot experiments shown in Figure 1 resemble standard small angle gradient echo sequences, with an additional quadratic encoding gradient to impose a quadratic phase on the slice. These are hybrid sequences with *k*-encoding in one direction and spatio-temporal encoding in the other. Figure 1(a) shows a hybrid of spatio-temporal
encoding and phase encoding, while Figure 1(b) shows a hybrid of spatio-temporal encoding and frequency encoding.





Although there are small differences between the results of these two sequences, $k$-encoding is performed with linear gradients in both sequences, while spatio-temporal encoding is achieved with quadratic encoding gradients. Ideally, one should have a pulsed quadratic encoding gradient that can be gated, i.e., turned off and on during the sequence, like any other pulsed field gradient. However, our experiments were performed with a continuous quadratic encoding gradient provided by shim coils. In the ideal case of Figure 1(a), where the quadratic encoding gradient is only applied during the interval $\tau$ between excitation and acquisition, the resulting phase profile (at time $t$ after starting signal acquisition) can be described by:

$$\varphi = \beta z^2 + k_z(t)z \tag{1}$$

where $\beta = 2\pi\gamma\alpha\tau$ is the quadratic term coefficient, $\gamma$ is the gyromagnetic ratio, $\alpha$ and $\tau$ are the amplitude of the quadratic encoding gradient and its duration respectively, $z$ is the coordinate along the spatio-temporal encoding direction, and $k_z(t)$ is $k$-space coordinate in this dimension at time $t$. This quadratic phase profile implies that the phase of the magnetization varies steeply as a function of the $z$ coordinate in all parts of the sample except near the vertex of the quadratic profile, where the variations of the phase are small (Paquin et al., 2009; Tal and Frydman, 2006), so that $d\varphi/dz \approx 0$. Under such conditions the signal originates almost exclusively from the vertex. When $k_z(t)$ changes, the position of the vertex also changes:

$$\frac{d\varphi}{dz} = 2\beta z + k_z(t) = 0 \Rightarrow z(t) = \frac{-k_z(t)}{2\beta} \tag{2}$$

thus the FOV in the spatio-temporal encoding dimension is:

$$FOV_{spatial\ encoding} = \frac{\Delta k_z}{2\beta} \tag{3}$$

where $\Delta k_z$ is the relevant range of $k$-space coordinates. As a result, the signal obtained by this method provides a map of the object in spatio-temporal encoding direction, so that there is no need for a Fourier transformation. Similar equations can be applied for the sequence of Figure 1(b), with the difference that $k_z$ is constant for each individual scan, although it varies between different scans. The signals obtained with the sequences of Figure 1 were then arranged in a 2D array, similar to other SPEN techniques, and a 1D Fourier transformation was performed in the $k$-encoding direction in order to obtain an image.

Ideally the quadratic encoding gradient in Figure 1 should be switched on after the excitation period and switched off before signal acquisition. In our experiments we have used a shim coil of our MRI apparatus to generate a quadratic encoding gradient, which is not designed to be switched in the course of a pulse program. Hence, in our experiments the quadratic encoding gradient is applied continuously. The existence of a quadratic encoding gradient in the excitation period only cause some minor de-phasing effects. However, during the signal acquisition a misregistration of the voxels in the spatio-temporal encoding direction (i.e., in the vertical direction in our images) will occur with respect to the outcome of Equation 2. In the absence of this voxel misregistration, successive data points in the spatially encoded signal correspond to equally-spaced points in the image (i.e., in real physical space). Misregistration occurs when successive data points are not equally spaced in the image. Nevertheless, a knowledge of the strength of the quadratic encoding gradient together with the ratio of the acquisition time and the evolution time allows for a correction of this misregistration. For multi-shot images this misregistration is very small (and can be neglected) because the acquisition time is much shorter than the evolution time. However, for single-shot

experiments it is not negligible, and has to be taken into account. Due to this misregistration, the real FOV (shown in Figures 4, 5, and 6) is smaller than it would be the ideal case when the quadratic encoding gradient can be switched on and off in the course of the sequence. Furthermore, the centre of the real FOV does not coincide anymore with the centre of our linear gradient coil, as can be seen in the single-shot images of Figure 6.

The pulse program used for the single-shot experiments is shown in Figure 2. This pulse program resembles an EPI
pulse program with an additional quadratic encoding gradient. Similar to multi-shot experiments, this quadratic encoding gradient should ideally be switched off during signal acquisition. Since in our experiments it extends throughout the acquisition interval, a non-negligible misregistration of the voxels in the spatio-temporal encoding direction (vertical direction in our images) occurs.

## 3 Method

Our experiments have been carried out on a Bruker 800 MHz NMR spectrometer equipped with a 25 mm micro-imaging probe and $x$, $y$ and $z$ pulsed field gradient amplifiers for micro-imaging. The phantom was a dumbbell-shaped piece of plastic that was placed in a 25 mm O.D. plastic tube filled with de-ionized water. The pulse programs were written and implemented using the Paravision program, and the data were processed either by a modified processing algorithm in Paravision or by a home-written Python code. The corrections needed to compensate for the above-mentioned misregistration were implemented using
a home-written Python code. In all cases, a sagittal slice (parallel to the $yz$ plane) with a thickness of about 0.7 mm was selected by using a linear gradient and a shaped *rf* pulse. In all images, the vertical $z$ axis is parallel to the main static field. Other parameters are given in the captions to the figures.

## 4 Results

Figure 3(a) shows a conventional multi-shot gradient echo reference image of our dumbbell-shaped phantom. In Figure 3(b) a
conventional gradient echo EPI image is depicted. Clearly the latter image is distorted due to field inhomogeneities caused by the phantom. In all experiments, the phantom extends beyond the vertical FOV. In Figures 3(a) and 3(b) however, no aliasing (no wrap-around artefacts) occur because, in these images the frequency encoding direction was chosen to be along the $z$ direction where the object extends beyond the FOV. As a consequence, signals stemming from areas located beyond the FOV are easily removed by digital filtering.
The top row in Figure 4 represents results obtained by the sequence shown in Figure 1(a), which is a multi-shot SPEN sequence. As is depicted in Figure 1(a), the quadratic phase profile required for spatio-temporal encoding is generated by a separate quadratic encoding gradient channel. In Figure 4, the amplitude of the quadratic encoding gradient is increasing from left to right, and as a result, the vertical FOV is decreasing from left to right, in accordance with Equation 2. Increasing the amplitude of the quadratic encoding gradient also results in better resolution, as will be explained in the Discussion section.





In the sequence of Figure 1(a), the quadratic encoding gradient can also be treated as a contribution to the inhomogeneity of the static field. Under such conditions, the usual theory of Fourier MRI is applicable, and one can apply a 2D Fourier transform to the raw data obtained from the sequence of Figure 1(a). These images are shown in the bottom row of Figure 4. Of course, in this case, the vertical FOV is not controlled by the amplitude of the quadratic encoding gradient, but is determined by the parameters that govern the FOV in Fourier MRI, e.g., the duration and the amplitude of the linear

gradients.

        The SPEN images obtained by the sequence shown in Figure 1(b), are depicted in the top row of Figure 5, in which the amplitude of the quadratic encoding gradient is increasing from left to right. The pulse sequences of Figures 1(a) and (b) are similar, the difference is that Figure 1(a) is a hybrid of spatio-temporal encoding and phase encoding, while Figure 1(b) is a hybrid of spatio-temporal encoding and frequency encoding. The top rows of Figure 4 and Figure 5 are similar, but their

bottom rows are not. Although the phantom extends beyond the vertical FOV, no aliasing is seen in Figure 4. In the top row of Figure 4, the $z$ dimension is spatially encoded, and no Fourier transformation is applied in that direction. As a result, aliasing which is a consequence of Fourier transformation cannot occur. Aliasing is also absent in the bottom row of Figure 4, despite the fact that those images are $k$-encoded in both directions, and a 2D Fourier transformation have been applied on the raw data. This is because, in those images the signals stemming from areas located beyond the FOV have been removed by digital

filtering. In the top row of Figure 5, no aliasing occurs, for the same reason as in the top row of Figure 4. However, in the images of the bottom row aliasing does occur, since the $z$ dimension is phase encoded. In the direction of phase encoding, the signal of the regions beyond the FOV are not filtered, and therefore, they fold back into the image.

        Four single-shot SPEN images obtained by the pulse sequence of Figure 2 are shown in Figure 6. The distortions in the images are much smaller than in the EPI image shown in Figure 3. This is a general property of single-shot SPEN methods,

which are much less sensitive to field inhomogeneities (and more generally to other sources of frequency variations like chemical shifts) than single-shot $k$-encoding methods. The quadratic encoding gradient is stronger in Figure 6(b) than in Figure 6(a). Consequently, the FOV is smaller, and the resolution is better. The amplitude of the quadratic encoding gradient is the same in Figures 6(d), 6(a), and 6(c), but the strength (peak amplitude) of the "blip" decoding gradients in Figure 2 has been increased respectively. Thus in 6(d) the strength of the blip decoding gradient is twice as small as in 6(a). In 6(c) the strength

of the blip decoding gradient is twice as large as in 6(a). As a result, the FOV in the spatio-temporal encoding direction of 6(d) is half of the FOV in 6(a), while in 6(c) it is twice as large as in 6(a). As explained in the Theory section, a misregistration occurs in Figure 6 due to the persistence of the quadratic encoding gradient throughout the detection interval. This effect has been compensated for Figure 6.

        Fig. 7 depicts four SPEN images obtained with the RASER (Chamberlain et al., 2007) sequence, with different spatio-

temporal encoding bandwidths that correspond to different amplitudes of the quadratic encoding gradients in Fig. 6. The improvement of the spatial resolution with increasing spatio-temporal encoding bandwidth (which is equivalent to increasing the amplitude of the quadratic encoding gradient) is evident in Fig. 7.





## 5 Discussion

In this work, we have shown that a quadratic encoding gradient can be used in spatio-temporal encoding methods. There are
several advantages of this method over traditional methods that rely on the simultaneous application of linear gradients and
frequency-modulated chirp pulses. In comparison to traditional methods, the new method allows one to reduce the SARs and
shorten the echo times (TEs). Also, it is easier to apply it for multi-slice applications, although our limitations did not allow to
explore these perspectives. To prove the validity of the theory, and to demonstrate some of its applications, several phenomena
have been investigated.

First, according to Equation 2, the FOV in the spatio-temporal encoding direction is a function of the amplitude of
the quadratic encoding gradient, as illustrated in Figures 4, 5, and 6. If the linear decoding gradients are kept unchanged, the
stronger the applied quadratic encoding gradient, the smaller the FOV.

The second point is the presence of aliasing in the phase encoding direction of traditional Fourier $k$-space images, in
contrast to SPEN images, as is evident from Figure 5.

Third, it is well known that, in analogy to traditional spatio-temporal encoding methods, the larger the quadratic phase
coefficient ($\beta$ in Equation 1), the better the resolution. This is normally achieved by using longer chirp pulses or pulses with
larger bandwidths, which are however associated with larger SARs. As can be seen in Figures 4, 5, and 6, the method
represented in this work allows one to achieve better resolution by increasing the strength of the quadratic encoding gradient,
without increasing the SAR.

There are some artefacts in our images, especially in those shown in Figure 6. The low quality of these images can be attributed
to imperfections of our quadratic encoding gradients. Ideally one should use dedicated gradient coils that produce a pure
quadratic encoding gradient. In this work however, a $z^2$ shim coil has been used to generate a gradient. The magnetic field
produced by an ideal $z^2$ shim coil is given by the following equation (Chmurny and Hoult, 1990).

$$B_z \propto 2z^2 - (x^2 + y^2) \tag{4}$$

If one selects a sagittal slice centred around a chosen value $x_0$, the $x^2$ term is replaced by a constant. The remaining $y^2$ term
however may lead to some artefacts in our images. It should be noted that Figures 4 and 5 can be considered to be Fourier
encoded in both directions, or Fourier encoded in one dimension and spatially encoded in the other. That is why the raw data
can be converted into an image either by a 1D or 2D Fourier transform.

## 6 Conclusions

In this work an alternative method has been presented to impart a quadratic phase onto the transverse magnetization
to be used for spatio-temporal encoding MRI. It allows one to reduce the specific absorption rate (SAR) and to shorten the
echo times (TE). Single- and multi-shot images have been obtained to validate the concept and to demonstrate the potential of
the method.



## 7 Author contributions

SM and PP conceived the experiments, SM conducted the experiments, SM and PP analyzed the results, SM, PP and GB wrote the paper. All authors have reviewed the manuscript.

## 8 Competing interests

The authors declare that they have no conflicts of interest.

## 9 Acknowledgments

The authors acknowledge financial support of the European Research Council (ERC, project 339754 "Dilute para-water"), the Equipex program "Paris en Résonance", and the support of GIS-IBISA program of the IMACHEM imaging platform (École Normale Supérieure – Collège de France).

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





**Figure 1: Multi-shot spatio-temporal encoding sequences used in this work. Slice selection and *k*-encoding are performed by linear gradients like in Fourier imaging sequences. Spatio-temporal encoding is achieved using a quadratic encoding gradient. Note that in contrast to conventional SPEN methods, no chirp pulse is used in this method.**






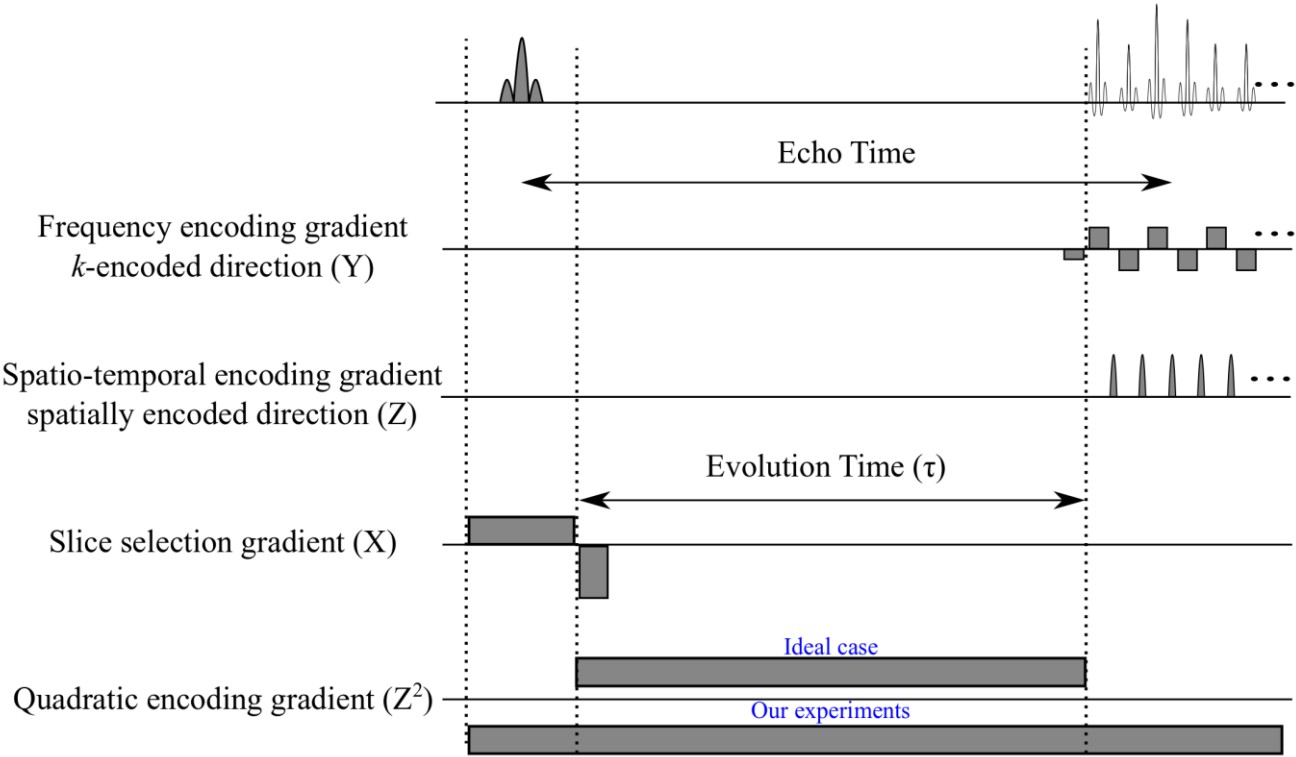

**Figure 2:** Single-shot spatio-temporal encoding sequence used in this work. Slice selection and k-encoding are performed by linear gradients similar to the EPI sequence. Spatio-temporal encoding is achieved using a quadratic encoding gradient. Note that, in contrast to conventional SPEN methods, no chirp pulse is used in this method.




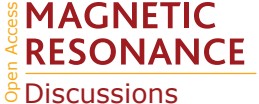

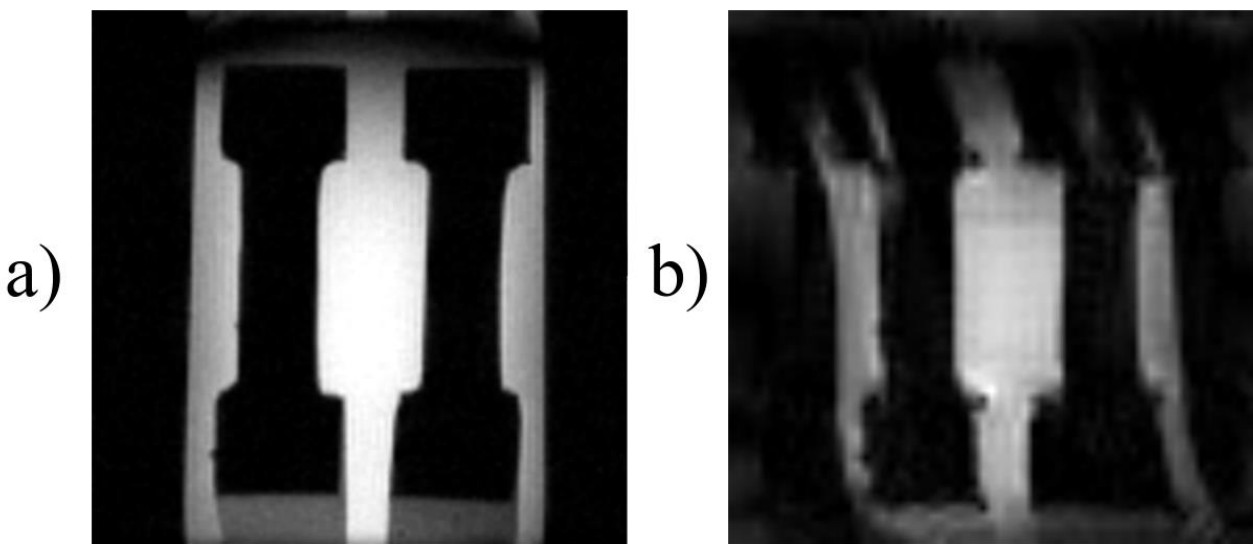

**Figure 3: (a) Reference image of our dumbbell-shaped phantom using only linear gradients with a standard gradient echo multi-shot sequence with the following parameters: matrix size 128 × 128, FOV = 32.0 × 32.0 mm, TE = 3.3 ms, flip angle = 30°, TR = 100 ms, readout bandwidth bw = 500 kHz. (b) A single-shot image of the phantom obtained with a standard gradient echo EPI sequence with the following parameters: matrix size = 64 × 64, FOV = 32.0 × 32.0 mm, TE = 7.2 ms, flip angle = 90°, readout bandwidth bw = 652 kHz. The slice thickness was about 0.7 mm in both cases.**






**Figure 4: Top row: three SPEN images obtained by the multi-shot sequence of Figure 1, using a quadratic encoding gradient, with the following parameters: matrix size 128 × 128, TE = 3.3 ms, flip angle 30°, TR = 100 ms, and readout bandwidth bw = 500 kHz. In (a) the amplitude of the quadratic encoding gradient was $\alpha \approx 0.22$ T.m$^{-2}$ and the FOV = 63.5 × 32.0 mm; in (b) $\alpha \approx 0.34$ T.m$^{-2}$ and the FOV = 42.3 × 32.0 mm; and in (c) $\alpha \approx 0.45$ T.m$^{-2}$ and FOV = 31.7 × 32.0 mm. These images are hybrids of spatio-temporal encoding (along the vertical z axis) and phase encoding (along the horizontal y axis). Increasing the amplitude of the quadratic encoding gradient will decrease the FOV in the spatio-temporal encoding direction. Bottom row: (d, e, and f) images obtained by applying a Fourier transformation also in the vertical direction (2D FT) on the same raw data as the top row. (d) Raw data of image (a); (e) raw data of image (b); (f) raw data of image (c). For all bottom row images the FOV = 32.0 × 32.0 mm.**





**Figure 5: Top row: three SPEN images obtained with the multi-shot sequence of Figure 1, using a quadratic encoding gradient, with the following parameters: matrix size 128 × 128, TE = 3.3 ms, flip angle 30°, TR = 100 ms, and readout bandwidth bw = 500 kHz. In (a) the amplitude of the quadratic encoding gradient was $\alpha \approx 0.22$ T.m$^{-2}$ and FOV = 63.5 × 32.0 mm; in (b) $\alpha \approx 0.34$ T.m$^{-2}$ and FOV = 42.3 × 32.0 mm; and in (c) $\alpha \approx 0.45$ T.m$^{-2}$ and FOV = 31.7 × 32.0. These images are hybrids of spatio-temporal encoding (along the vertical z axis) and frequency encoding (along the horizontal y axis). Increasing the amplitude of the quadratic encoding gradient will decrease the FOV in the spatio-temporal encoding direction. Bottom row: (d, e, and f) images obtained by applying a Fourier transformation also in the vertical direction (2D FT) on the same raw data as the top row. (d) Raw data of image (a); (e) raw data of image (b); (f) raw data of image (c). For all bottom row images the FOV = 32.0 × 32.0. Aliasing is evident in three images of the bottom row.**



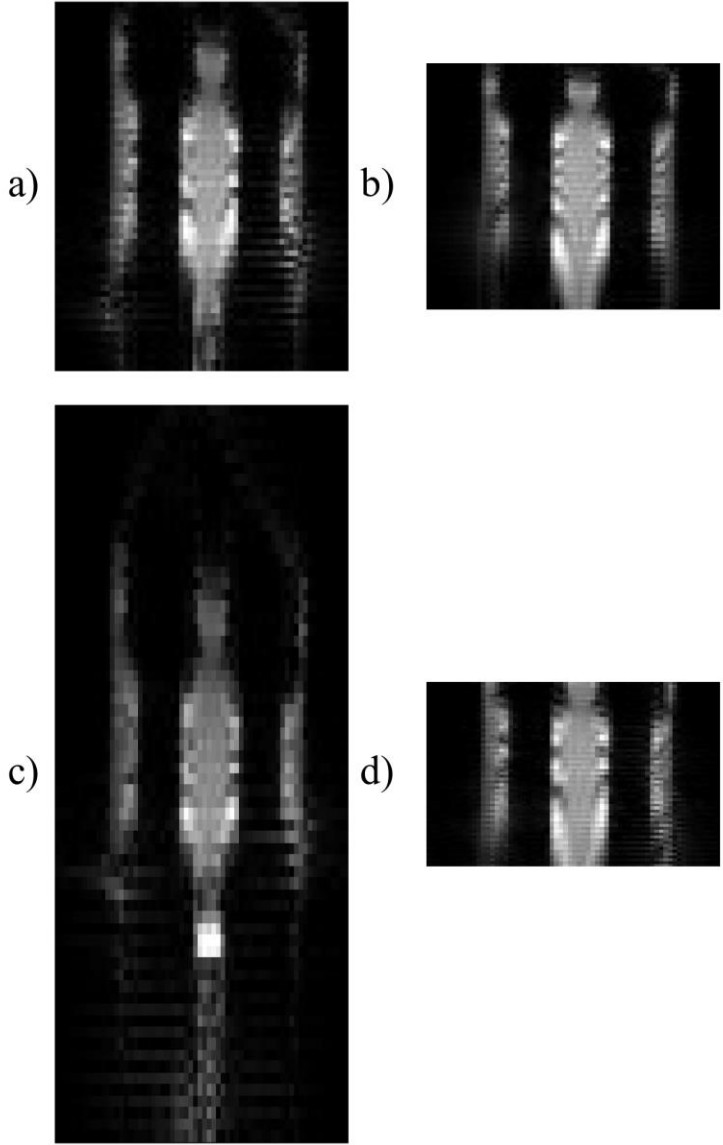

**Figure 6: Single-shot SPEN images obtained with the sequence shown in Figure 2, using a quadratic encoding gradient, with the**
**following parameters: matrix size 64 × 64, TE = 7.2 ms, flip angle 90˚, evolution time $\tau$ = 2.66 ms, and readout bandwidth bw = 652**
**kHz. In (a) the amplitude of the quadratic encoding gradient pulse was $\alpha \approx 0.13$ T.m$^{-2}$ (corresponding to a quadratic phase coefficient**
**$\beta \approx 3.33$ cm$^{-2}$), the peak amplitude of the linear decoding gradient (blip gradients) $G_{decoding}$ = 0.014 T.m$^{-1}$, and FOV = 40.3 × 32.0**
**mm; in (b) $\alpha \approx 0.20$ T.m$^{-2}$ (corresponding to $\beta \approx 4.99$ cm$^{-2}$), $G_{decoding}$ = 0.014 T.m$^{-1}$, and FOV = 26.9 × 32.0 mm; in (c) $\alpha \approx 0.13$ T.m$^{-2}$**
**(corresponding to $\beta \approx 3.33$ cm$^{-2}$), $G_{decoding}$ = 0.028 T.m$^{-1}$, and FOV = 80.6 × 32.0 mm; in (d) $\alpha \approx 0.13$ T.m$^{-2}$ (corresponding to $\beta \approx 3.33$**
**cm$^{-2}$), $G_{decoding}$ = 0.007 T m$^{-1}$, and FOV = 20.1 × 32.0 mm. These images are hybrids of spatio-temporal encoding (along the vertical**
**z axis) and frequency encoding (along the horizontal y axis). Increasing the amplitude of the quadratic encoding gradient will**
**decrease the FOV in the spatio-temporal encoding direction. Conversely, increasing the amplitude of the linear decoding gradient**
**will increase the FOV in the spatio-temporal encoding direction. Note that for single-shot images there is a non-negligible**
**misregistration of the voxels. The images have been corrected to compensate for this effect.**






Figure 7: **Traditional single-shot SPEN images obtained by the RASER**(Chamberlain et al., 2007) **sequence using a chirp pulse and linear gradients, with the following parameters: matrix size 64 × 64, TE = 37.8 ms, FOV = 32.0 × 32.0 mm, flip angle 90°, and readout**
**bandwidth 652 kHz. The spatio-temporal encoding bandwidth (bw$_{se}$) was decreased stepwise: (a) bw$_{se}$ = 31.0 kHz (corresponding to a quadratic phase coefficient $\beta \approx 13.77$ cm$^{-2}$); (b) bw$_{se}$ = 15.5 kHz (corresponding to $\beta \approx 6.89$ cm$^{-2}$; (c) bw$_{se}$ = 7.8 kHz (corresponding to $\beta \approx 3.44$ cm$^{-2}$); and (d) bw$_{se}$ = 3.9 kHz (corresponding to $\beta \approx 1.72$ cm$^{-2}$). These images are hybrids of spatio-temporal encoding (along the vertical z axis parallel to the static field) and frequency encoding (along the horizontal y axis).**

