# Peer review of "Spatio-temporal encoding by quadratic gradients in magnetic resonance imaging"

_Magnetic Resonance, 2019_

## Referee Comment (RC1) · Anonymous Referee #1 · 26 Nov 2019

mr-2019-2, Marhabaie et al

The paper present a method of SPEN imaging that replaces the frequency-swept CHIRP pulse with a quadratic gradient, claiming advantages in SAR and echo time. Results of quadratic gradient SPEN are shown from a phantom.

No quantitative comparisons of image metrics (including SNR, resolution artefacts etc) are provided vs standard methods: chirped SPEN, or conventional MRI. Thus the conclusions concerning advantages are unsubstantiated. Significant artefacts associated with the encoding are evident but the dependence of these on practical instrumental factors are not elaborated. The authors do not provide a compelling case for using the proposed method vs existing SPEN or regular MRI methods. Moving the author's approach forward would basically require new sets of comparative studies and analyses.

[Figure]

1. Abstract. (i)line 12. "In this work, we show that it can be advantageous..." Advantageous compared to what? (ii)No quantitative comparison of performance metrics are documented or summarized. SAR and TE are mentioned–these only relate to the RF chirp pulse. What about the whole new gradient system required? "Resolution, FOV, SNR are the same" how was this tested? (iii)Why are quadratic gradients–which are virtually non-existent for spatial encoding in NMR systems, advantageous compared to linear gradient systems that virtually all NMR imaging systems have?

2. Introduction p1 line 21. If the SPEN method requires "sequential excitation" and "sequential detection", it is surely at a disadvantage in MRI efficiency (SNR per unit time) to conventional MRI methods wherein certain multi-dimensional encoding can be performed concurrently. Same would be true of the dephasing effect. Is this correct or should the text be better clarified?

3. Introduction p2 lines 36-50. It seems that the authors are basically exchanging the high SAR/RF problems required to swamp out the phase variations due to Bo inhomogeneity, for problems with generating quadratic gradients which aren't acknowledged.

4. Introduction, last para. Please summarize what you plan to show in this paper.

5. Theory. Fig 1. (i)The presence of the quadratic Z gradient during regular slice selection in the x-direction will degrade SNR, slice selection and distort the imaging plane excited in a way that varies quadratically along the Z-axis. This is not "sequential excitation". The same appears true for the y-direction with linear encoding gradients and $Z^2$ encoding running currently. (ii)The caption says "...linear gradients like in Fourier imaging sequences." This appears to contradict the statement "there is no need for a Fourier transformation in the spatio-temporal encoding direction" (p1). The authors have both phase-encoding gradient steps and a quadratic gradient in the same dimension. (Why is this advanatgeous to not applying the quadratic gradient?)

6. Theory p3 line 63, "...while spatio-temporal encoding is achieved with quadratic encoding gradients". I have an issue with the term "spatio-temporal encoding (direc-
regular MRI sequence employing linear gradients. Without that the work is anecdotal.

12. Results p5, Line 149, Fig 7. What is the RASER sequence? Why is this being compared? (Is this the best SPEN sequence that there is?) It is a major detriment to the paper that no comparison with a CHIRP pulse sequence is mentioned until one arrives at the caption of figure 7. But there are still no quantitative comparisons here, and the results seem to show that neither SPEN sequences are suitable for MRI (vs. Fig 3a, say).

13. Discussion, p6. (i)line 155. The authors have not documented any advantages of their method. (ii)They have not characterized the theoretical dependence of image performance metrics on the gradient system properties which would be needed for practical implementation. They attribute artefacts (eg in Fig 6) to such limitations, but basically it is arguable looking at the qualitative results, that the method just exchanges an SAR problem (which would probably not exist for their small-bore system) with CHIRP pulses for a reduction in image quality and performance, but which are still inferior to existing MRI methods.

---

## Referee Comment (RC2) · Anonymous Referee #2 · 19 Dec 2019

Summary: This paper describes a method to accomplish spatiotemporal-encoded (SPEN) MRI using a quadratic field gradient. Although this work shows qualitatively that this approach is feasible, the quality of the images shown are not particularly impressive, and the lack of any quantitative comparisons with conventional SPEN MRI, leave much to be desired. In addition, the proposed method is not novel since others have used quadratic encoding gradients in MRI.

Specific comments:

1. Line 54-55: "quadratic encoding gradients have not yet been applied to spatio-temporal encoding (SPEN) methods that normally use chirp pulses". A more accurate statement might be: The spatiotemporal dependence of MR signals encoded with non-linear gradients was previously noted (eg, see Zaitsev et al, Magn Reson Med
73:1407–1419, 2015), although that work did not explicitly describe the phenomenon in the context of SPEN.

2. Eqn 3: I believe the intended meaning of delta_k here is different from the conventional delta_k used in describing Fourier-encoded MRI, in which 1/delta_k specifies the FOV. To avoid confusion, the authors might want to make this difference clear or use a different variable. The current description of delta_k, "relevant range of k-space coordinate", is a bit ambiguous.

3. Figs 4a-c, 5a-c, and 6a-d: There are very noticeable artifacts in the SPEN images. The authors should comment on the potential sources of these.

4. Figs 4d-f and 5d-f: These images are distorted due to the background quadratic field, but it is difficult to assess the distortion without a reference image nearby.

5. Methods and Fig 7: The authors should specify which version of RASER was used and provide more of the relevant parameters. What was the duration of the chirp pulses? Were they standard chirp pulses or some variant like HS20? Was a blipped or continuous gradient used in the spatiotemporal-encoded direction? If the latter, was the phase correction (Eqn 1) provided in the paper by Chamberlain et al used? The vertical banding and jaggedness of the vertical edges of the object suggests the post-processing procedures were not optimal.

---

## Author Comment (AC1) · 30 Jan 2020

Département de Chimie
Ecole Normale Supérieure
24 Rue Lhomond, Paris 75005, France
sina.marhabaie@ens.fr

[Figure]

To Dr. Paul Bottomley,
Associate Editor of MR
January 30th 2020

Re: Magn. Reson. Discuss. https://doi.org/10.5194/mr-2019-2-RC2, 2019
Spatio-temporal encoding by quadratic gradients in magnetic resonance imaging
Sina Marhabaie, Geoffrey Bodenhausen, Philippe Pelupessy

Dear Dr. Bottomley,

We have received comments from two anonymous referees, reproduced below in Roman. Our answers to the anonymous referees are inserted *in italics*. We have revised the paper. In the revised manuscript, our modifications are highlighted in yellow. With current hardware, we cannot improve the images, nor can we provide meaningful comparisons. We hope that you will find this work acceptable as a proof-of-concept.

Best wishes

Sina Marhabaie

**Anonymous Referee #1**

mr-2019-2, Marhabaie et al

The paper present a method of SPEN imaging that replaces the frequency-swept CHIRP pulse with a quadratic gradient, claiming advantages in SAR and echo time. Results of quadratic gradient SPEN are shown from a phantom.

*This summary of our paper is correct.*

No quantitative comparisons of image metrics (including SNR, resolution artefacts etc) are provided vs standard methods: chirped SPEN, or conventional MRI. Thus the conclusions concerning advantages are unsubstantiated.

*Although our instrument can record images with either linear or quadratic shim gradients, the gradient coils are different. The former can be switched while the latter cannot. The former are close to ideal (nearly pure $G_z$) while the latter are not pure since they are of the form $G_z^2 - (G_y^2 + G_x^2)/2$. It is therefore difficult to make meaningful comparisons. This work is merely intended as a proof of concept, in the hope of motivating new instrumental and conceptual developments. In addition, Fig. 7 shows four images obtained with chirped SPEN that can be used for qualitative comparison.*

Significant artefacts associated with the encoding are evident but the dependence of these on practical instrumental factors are not elaborated. The authors do not provide a compelling case for using the proposed method vs existing SPEN or regular MRI methods. Moving the author's approach forward would basically require new sets of comparative studies and analyses.

1. Abstract. (i)line 12. "In this work, we show that it can be advantageous. . ." Advantageous compared to what?

*Compared to the combination of a linear gradient and a chirp pulse, one can achieve much smaller specific absorption rates (SARs) since no rf pulses are needed to create a quadratic phase profile. Slice selection (or sequential multiple slice excitation) is much easier since there is no need for a CHIRP pulse that touches all spins in the object. These advantages are inherent to our new method.*

(ii) No quantitative comparison of performance metrics are documented or summarized. SAR and TE are mentioned–these only relate to the RF chirp pulse. What about the whole new gradient system required?

*As stated in our paper, we have use quadratic gradient coils designed to shim the static field, rather than the linear gradients that are an integral part of the MRI accessory. This is far from ideal. Only future will tell if the new strategy will stimulate the development of new hardware.*

"Resolution, FOV, SNR are the same" how was this tested? (iii) Why are quadratic gradients– which are virtually non-existent for spatial encoding in NMR systems, advantageous compared to linear gradient systems that virtually all NMR imaging systems have?

*Commercial imaging systems are not equipped with coils to generate switched quadratic gradients. If our method were to be widely accepted by the community, the instrument manufacturers would have to make a significant investment by designing new MRI accessories with integrated quadratic gradient coils.*

2. Introduction p1 line 21. If the SPEN method requires "sequential excitation" and "sequential detection", it is surely at a disadvantage in MRI efficiency (SNR per unit time) to conventional MRI methods wherein certain multi-dimensional encoding can be performed concurrently. Same would be true of the dephasing effect. Is this correct or should the text be better clarified?

*Because it relies on "sequential excitation" and "sequential detection", SPEN is less efficient than conventional FT MRI methods. Our suggestion to modify SPEN by replacing linear gradients by quadratic gradients does not modify this inherent disadvantage of SPEN. A comparison between SPEN and FT-MRI is beyond the scope of our paper, and has already made elsewhere.*

3. Introduction p2 lines 36-50. It seems that the authors are basically exchanging the high SAR/RF problems required to swamp out the phase variations due to Bo inhomogeneity, for problems with generating quadratic gradients which aren't acknowledged.

*Our method significantly reduces SAR when compared to current SPEN methods. However, the challenge of generating high-quality switched quadratic gradients is a matter of engineering that can be resolved, unlike the issue of SAR associated with conventional SPEN.*

4. Introduction, last para. Please summarize what you plan to show in this paper.

*We have inserted a brief paragraph at the end of the introduction.*

5. Theory. Fig 1. (i) The presence of the quadratic Z gradient during regular slice selection in the x-direction will degrade SNR, slice selection and distort the imaging plane excited in a way that varies quadratically along the Z-axis. This is not "sequential excitation". The same appears true for the y-direction with linear encoding gradients and $Z^2$ encoding running currently. (ii) The caption says ". . . linear gradients like in Fourier imaging sequences." This appears to contradict the statement "there is no need for a Fourier transformation in the spatio-temporal encoding direction" (p1). The authors have both phase-encoding gradient steps and a quadratic

gradient in the same dimension. (Why is this advantageous to not applying the quadratic gradient?)

*(i): During slice selection with a $G_x$ gradient, there is also a weak $G_z^2$ gradient, because our instrumentation is not designed for rapid switching of the shim coils. The very fact that the $G_z^2$ gradient is much weaker that the $G_x$ gradient allows us to neglect this effect. With instrumentation for rapid switching, this problem disappears altogether.*
*(ii): Our imaging sequences are spatially encoded only in one dimension. In the second dimension they are k-encoded like any other ordinary imaging sequence. Therefore, to produce a two dimensional image, we need to apply a one dimensional Fourier transformation in the k-encoded dimension (as mentioned in the manuscript, as long as the Nyquist theorem is fulfilled—in our case only for the multi-shot images—one can still apply FT on both directions). Furthermore, the merits of SPEN (chirped SPEN or our method) appear mostly in single-shot imaging. The multi-shot sequences in this work were designed to prove the concept of using quadratic gradients instead of chirp pulses.*

6. Theory p3 line 63, "while spatio-temporal encoding is achieved with quadratic encoding gradients". I have an issue with the term "spatio-temporal encoding (direction)" as some kind of differentiator vs. conventional spatial encoding in MRI. All conventional MRI methods encode in the time domain (which is equivalent to k-space). Phase-encoding cause position dependent changes in the time (temporal) domain. So does the regular read-out gradient, and slice selection. That is, in Fig 1 spatio-temporal encoding is achieved with all of the gradients in all dimensions, not just the quadratic one.

*Although "spatio-temporal encoding" is a somewhat unfortunate expression, it is currently widely used in the SPEN literature. In the MRI literature the word "spatial encoding" has been used to refer to different concepts, and has different meanings. One of these meanings, also known as time encoding, is an encoding method where spatial information is directly encoded in the amplitude of the signal. We have used the word "spatio-temporal encoding" to refer to all types of methods under a single name.*

7. Theory p3. (i) Not turning the gradients off when encoding with other gradients is a serious deficiency for a "proof of principle demonstration". This means someone would have to come along and implement the method properly and redo any comparisons in another paper on the same idea–basically a redo. (ii) The method of correcting the "non-negligible" misregistration error" is not documented.

*Indeed, our method will need to be re-visited in due course, using an instrument that is capable of switching quadratic gradients. The method of correcting the non-negligible misregistration is now explained in the appendix of our revised paper.*

8. Theory. There are no theoretical analyses concerning the practical limits and requirements of the quadratic gradients: strength, fidelity and their relationship between to spatial resolution, spatial distortion, bandwidth per point, SNR per point, the potential variation of SNR and resolution with position from "the vertex", and system dynamic range.

*The theory for our method is the same as for "traditional" SPEN methods. This paper is not intended to review this theory.*

9. Method section, p4. The quadratic gradient is not specified. The image metrics being measured are unstated. No comparisons are specified. In particular, improvements are being claimed vs the chirp pulse method, but no such comparison is apparently being performed (according to the methods). Therefore the claims of advantage cannot be substantiated.

*We replaced to the methods section the sentence: "Other parameters are given in the captions to the figures." by "The experimental parameters like the strength of the quadratic gradients, their duration, etc. are given in the figure captions." Comparisons are not very meaningful for the reasons stated above.*

10. Method. I expect the use of the Bruker shim coil is not optimal for quadratic encoding but no information is provided concerning its fidelity over the imaging volume.

*Indeed, the Bruker shim coil is not optimal for quadratic encoding, but its fidelity appears to be sufficient.*

11. Results. It is problematic that no image performance metrics (SNR, resolution distortion, artefacts etc) are being provided or compared in any quantitative fashion. The appropriate comparisons here are the author's new method with quadratic gradients vs. (a) the CHIRP pulse against which the authors are claiming advantage; and (b) a regular MRI sequence employing linear gradients. Without that the work is anecdotal.

*Until the instrument manufacturers make a significant investment by designing new MRI accessories with integrated quadratic gradient coils, such comparisons are not meaningful. We believe that our work shows a novel avenue that has not been explored before.*

*Furthermore, a qualitative comparison has already been made in the article. Fig. 3(b) shows a single-shot k-encoded image, which is distorted due to magnetic field inhomogeneities. Fig. 7 show some chirped SPEN images recorded in the same conditions. These images are much less distorted, but they are associated to a very large SAR. Fig. 6 shows some images recorded with our method. These images are much less distorted than in Fig. 3(b), also their SAR is much less than in Fig. 7.*

12. Results p5, Line 149, Fig 7. What is the RASER sequence? Why is this being compared? (Is this the best SPEN sequence that there is?) It is a major detriment to the paper that no comparison with a CHIRP pulse sequence is mentioned until one arrives at the caption of figure 7. But there are still no quantitative comparisons here, and the results seem to show that neither SPEN sequences are suitable for MRI (vs. Fig 3a, say).

*We have explained in a few words that RASER is a special form of SPEN designed to compensate for variations in $T_2$ contrast due to sequential excitation. As stated above, due to hardware limitations we cannot make meaningful quantitative comparisons. Fig. 3(a) is a multi-shot image. None of the existing single-shot imaging techniques (k-encoded or SPEN) can produce an image that can compete a multi-shot image in the same conditions.*

13. Discussion, p6. (i) line 155. The authors have not documented any advantages of their method. (ii) They have not characterized the theoretical dependence of image performance metrics on the gradient system properties which would be needed for practical implementation. They attribute artefacts (eg in Fig 6) to such limitations, but basically it is arguable looking at the qualitative results, that the method just exchanges an SAR problem (which would probably not exist for their small-bore system) with CHIRP pulses for a reduction in image quality and performance, but which are still inferior to existing MRI methods.

*Comparisons have not been made at this time for the reasons stated above.*

**Anonymous Referee #2**

Summary: This paper describes a method to accomplish spatiotemporal-encoded (SPEN) MRI using a quadratic field gradient. Although this work shows qualitatively that this approach is feasible, the quality of the images shown are not particularly impressive, and the lack of any quantitative comparisons with conventional SPEN MRI, leave much to be desired. In addition, the proposed method is not novel since others have used quadratic encoding gradients in MRI.

*Indeed, quadratic encoding gradients have been used in MRI, but never with SPEN, to the best of our knowledge.*

Specific comments:

1. Line 54-55: "quadratic encoding gradients have not yet been applied to spatio-temporal encoding (SPEN) methods that normally use chirp pulses". A more accurate statement might be: The spatiotemporal dependence of MR signals encoded with non-linear gradients was previously noted (eg, see Zaitsev et al, Magn Reson Med 73:1407–1419, 2015), although that work did not explicitly describe the phenomenon in the context of SPEN.

*We have in inserted a reference to the work by Zaitsev, although it is not concerned with SPEN.*

2. Eqn 3: I believe the intended meaning of delta_k here is different from the conventional delta_k used in describing Fourier-encoded MRI, in which 1/delta_k specifies the FOV. To avoid confusion, the authors might want to make this difference clear or use a different variable. The current description of delta_k, "relevant range of $k$-space coordinate", is a bit ambiguous.

*In equations 1-3 the variable k has exactly the same definition as in conventional k used in Fourier-encoded MRI, in which 1/delta_k specifies the FOV of a Fourier-encoded image. The difference between Fourier-encoded MRI and spatially encoded MRI is the relation between k and FOV. In Fourier encoded MRI, the larger the delta_k the smaller the FOV, but in spatially encoded MRI the larger the delta_k the larger the FOV. In order to prevent any ambiguity we have replaced our k by $k^{se}$ (se = spatial encoding.)*

3. Figs 4a-c, 5a-c, and 6a-d: There are very noticeable artifacts in the SPEN images. The authors should comment on the potential sources of these.

*We attribute the distortions to the fact to different factors: (i) the fact that the quadratic gradients cannot be switched; (ii) that the quadratic gradients are not purely along one axis (i.e., that $G_z^2$ is contaminated with $(G_y^2 + G_x^2)$); (iii) the weakness of the quadratic shim gradients.*

4. Figs 4d-f and 5d-f: These images are distorted due to the background quadratic field, but it is difficult to assess the distortion without a reference image nearby.

*Although these images contain artefacts (as discussed above) they are much closer to the reference image, which is shown in Fig. 3(a) compared to single-shot Fourier techniques like EPI. An example of the latter is shown in Fig. 3(b).*

5. Methods and Fig 7: The authors should specify which version of RASER was used and provide more of the relevant parameters. What was the duration of the chirp pulses? Were they standard chirp pulses or some variant like HS20? Was a blipped or continuous gradient used in the spatiotemporal-encoded direction? If the latter, was the phase correction (Eqn 1) provided in the paper by Chamberlain et al used? The vertical banding and jaggedness of the vertical edges of the object suggests the post-processing procedures were not optimal.

*The requested information has been inserted in the caption of Fig. 7."Traditional single-shot SPEN images obtained by the RASER (Chamberlain et al., 2007) sequence using a "WURST" (Kupce and Freeman, 1995) chirp pulse and continuous linear gradient in the spatiotemporal-encoded direction, with the following parameters: matrix size 64 × 64, TE = 37.8 ms, FOV = 32.0 × 32.0 mm, flip angle 90˚, chirp pulse duration = 9.08 ms, and readout bandwidth 652 kHz. The spatio-temporal encoding bandwidth ($bw_{se}$) was decreased stepwise: (a) $bw_{se}$ = 31.0 kHz (corresponding to a quadratic phase coefficient $\beta \approx 13.77$ $cm^{-2}$); (b) $bw_{se}$ = 15.5 kHz (corresponding to $\beta \approx 6.89$ $cm^{-2}$; (c) $bw_{se}$ = 7.8 kHz (corresponding to $\beta \approx 3.44$ $cm^{-2}$); and (d) $bw_{se}$ = 3.9 kHz (corresponding to $\beta \approx 1.72$ $cm^{-2}$). These images are hybrids of spatio-temporal encoding (along the vertical z axis parallel to the static field) and frequency encoding (along the horizontal y axis)."*

*Residual vertical banding and jaggedness are not due to imperfect phase correction. We have used Chamberlain's prescriptions, which has proved to give nice results if one uses larger time-spatial encoding bandwidth, but is insufficient in our case.*